# Children and Women’s Health in South East Asia: Gap Analysis and Solutions

**DOI:** 10.3390/ijerph17103366

**Published:** 2020-05-12

**Authors:** Viroj Tangcharoensathien, Kunihiko Chris Hirabayashi, Chompoonut Topothai, Shaheda Viriyathorn, Orana Chandrasiri, Walaiporn Patcharanarumol

**Affiliations:** 1International Health Policy Program, Ministry of Public Health, Nonthaburi 11000, Thailand; viroj@ihpp.thaigov.net (V.T.); chompoonut@ihpp.thaigov.net (C.T.); shaheda@ihpp.thaigov.net (S.V.); orana@ihpp.thaigov.net (O.C.); walaiporn@ihpp.thaigov.net (W.P.); 2Health and HIV Section, United Nation Children’s Funds, Regional Office for East Asia and Pacific, Bangkok 10200, Thailand

**Keywords:** children health, women health, maternal and child health, East-Asia Pacific, universal health coverage

## Abstract

In response to the Millennium Development Goals (MDGs) and Sustainable Development Goals (SDGs) commitment, eight selected countries in the South East Asia region have made a remarkable reduction in infant and child mortality, while a few have achieved an SDG 3.2 target of 25 and 12 for child and neonatal mortality rate, respectively, well before 2030. Across these eight countries, there is a large variation in the achievement of the nine dimensions of maternal, neonatal, and child health service coverage. The poorest wealth quintiles who reside in rural areas are the most vulnerable and left behind from access to service. The rich rural residents are better off than the poor counterparts as they have financial means for travel and access to health services in urban town. The recent 2019 global Universal Health Coverage (UHC) monitoring produced a UHC service coverage index and an incidence of catastrophic health spending, which classified countries into four quadrants using global average. Countries belonging to a high coverage index and a low incidence of catastrophic spending are good performers. Countries having high coverage but also a high incidence of catastrophic spending need to improve their financial risk protection. Countries having low coverage and a high incidence of catastrophic spending need to boost service provision capacity, as well as expand financial protection. Countries having low coverage and a low incidence of catastrophic spending are the poor performers where both coverage and financial protection need significant improvement. In these countries, poor households who cannot afford to pay for health services may forego required care and instead choose to die at home. This paper recommended countries to spend adequately in the health sector, strengthen primary health care (PHC) and safeguard the poor, mothers and children as a priority in pathways towards UHC.

## 1. Introduction

At the United Nations General Assembly (UNGA) high level meeting on universal health coverage (UHC) in September 2019, a political declaration was adopted that re-enforces commitment to prior UNGA resolutions on Universal Health Coverage (UHC) contained in Sustainable Development Goals (SDGs) target 3.8. The Declaration calls for health, finance and other ministers to support progressive realization of UHC.

The survival, health and wellbeing of women, children and adolescents are essential to ending extreme poverty, promoting development and resilience, and achieving the SDGs. UHC is a major driver that contributes to the achievement of health-related SDG targets that synergistically contribute to maternal, newborn, and child health (MNCH) and wellbeing. UN member states need to accelerate the progress of UHC, which in turn contributes to children and women’s health—one of the best investments [1].

In addition, implementing the Global Strategy for Women’s, Children’s and Adolescents’ Health (2016–2030) [2] by increasing and sustaining financing would yield tremendous returns by 2030, such as ending preventable maternal, newborn, child and adolescent deaths and stillbirths [3,4]. Evidence shows that at least a 10-fold return on investments in the health and nutrition of women, children and adolescents [5,6]. The cognitive capital, nurtured at prenatal and early childhood periods, contributes to human capabilities [7], where governments need to support caregivers to nurture cognitive capital [8].

Based on published data from World Development Indicators, statistics from Multi-Indicator Cluster Survey (MICS) and Demographic Health Survey (DHS), this paper reviewed progress, analyzed inequity gaps in MNCH status and service coverage in eight selected countries in the South East Asia region. It also reviewed the UHC status in relation to MNCH health services coverage in each country and recommended policy solution. The selection of these eight countries was based on four criteria. Firstly, availability of the most updated, comparable and disaggregated statistics of MNCH by national household surveys either MICS or DHS. Secondly, countries have similar socio-cultural background. Thirdly, countries belong to the lower- and upper-middle income country group. Finally, countries have different levels of UHC that facilitate cross-country analysis of how UHC improves access to MNCH services. 

Descriptive data analysis was provided, spider webs presentation on the achievement of nine dimensions of MNCH service coverage was produced and compared across eight countries.

## 2. Countries at a Glance

A large variation is noted in Table 1 though all countries belong to lower middle-income countries except Thailand, an upper-middle income country. GDP growth is favorable in all except Timor Leste and Thailand. Non-tax revenue in Myanmar and Timor Leste is higher than half of the total revenue, while tax revenue is a smaller portion in GDP.

Fiscal space for health is measured by current health expenditure (CHE) per capita and as % GDP. Lao People’s Democratic Republic (Lao PDR) spent the least, Lao PDR, Myanmar and Cambodia spent less than US $80 per capita. 

Donor funding for health as % of CHE plays role in Timor Leste (31.6%), Cambodia (18.9%) and Lao PDR (18.1%), while Indonesia and Thailand are almost independent from donor funding, 0.4% and 0.2% of CHE.

The low level of domestic general government spending on health (GGHE) as % CHE in Cambodia (21.8%) and Myanmar (20.1%) results in high level of out-of-pocket payment (58.6% and 74.0%, respectively). The high level of out-of-pocket payment can cause catastrophic health expenditure to the households or the unaffordable medical bills prevent the poor from accessing essential services.

Although Timor Leste has a low level of out-of-pocket payment, 8.9% of current health expenditure, the domestic general government spending on health was also low, 3.2% of general government expenditure, and relies more on external funding, 31.6% of current health spending. The low level of out-of-pocket payment may reflect inadequate access to services.

In contrast, there is a high level of general government spending on health in Thailand, 78.1% of current health expenditure, and general government health expenditure is high, 15.3% of general government expenditure; both of these factors contribute to the low level of out-of-pocket payment, 12.1% of current health expenditure.

## 3. Protection of Children and Women: Achievement and Gaps

### 3.1. Health Status: Child, Infant, and Neonatal Mortality 

Countries had committed in the SDG 3.2 target, by 2030, to end preventable deaths of newborns and children under 5 years of age, with all countries aiming to reduce neonatal mortality (NMR) to at least as low as 12 per 1000 live births and under-5 mortality (U5MR) to at least as low as 25 per 1000 live births.

Across these eight selected countries, see Figure 1, Lao PDR, Myanmar and Timor Leste having U5MR more than 40 and NMR was 20 or more, there is a need to accelerate significant progress towards the achievement of 25 and 12 per 1000 live births by 2030. Thailand had achieved the SDG 3.2 targets by 2016. 

A long series of U5MR and infant mortality rate (IMR) in 1990 and 2016 during the Millennium Development Goals (MDGs) era, in Figure 2, showed that these countries had achieved hugely on child survival, particularly Timor Leste, Lao PDR, Myanmar and Cambodia. Such historical achievements through child survival programs proved that SDG commitment is also possible, if governments sustain their commitment and accelerate UHC progress. During the last decade between 2007 and 2017, the declines in neonatal mortality were less steep than child mortality, reflecting the needs for strengthening quality perinatal care. 

### 3.2. Maternal, Newborn, and Child Health Service Coverage: Recent Achievements and Challenges

The status of MNCH service coverage of eight countries were analyzed based on data obtained from the national household survey of each country either from MICS or DHS. The details of data collection and processing information can be found at https://mics.unicef.org/tools#survey-design [12] and https://dhsprogram.com/data/data-collection.cfm [13].

Figure 3a–h shows the spider web of eight selected countries on the nine dimensions of MNCH interventions: (a) family planning (modern methods); (b) four quality antenatal care (ANC) visits; (c) skilled birth attendant rates; (d) the early initiation of breastfeeding; (e) third dose of Diphtheria-tetanus-pertussis (DTP3) immunization coverage; (f) care seeking for children with pneumonia; (g) the use of improved drinking water sources; (h) the use of improved sanitation; and coverage of birth registration.

Thailand (Figure 3a), had achieved a high level of coverage, eight out of nine dimensions of MNCH interventions, except for a low level of early initiation of breastfeeding, as a result of the aggressive marketing of breast-milk substitute industries. A study on the violations of the 1981 International Code of Marketing of Breast-milk Substitutes found that Thailand had the highest rate of pregnant women given formula milk samples [15]. Breast-milk substitute industries used hospitals, health centers, private clinics, and pharmacies as market promotion platforms, by giving samples of infant and young child food as gifts to mothers or health personnel, providing information on infant and young child food products and having sale persons contact pregnant women and mothers directly. Yet, Thailand has to scale up the early initiation of breastfeeding through strengthening the Baby-Friendly Hospital Initiative [16].

Thailand had decided to legislate the voluntary code into national law. This took the form of the Control of Marketing Promotion of Infant and Young Child Food Act B.E.2560 in 2017. During the legislative process, there was strong resistance from certain pediatricians and the Thai Medical Council who amplified the downsides of breastfeeding by quoting a study in Nepal [17] that “prolonged breastfeeding beyond 12 months results in stunting”. This concealed the multiple factors contributing to stunting, such as socio-economic status, maternal education, poverty, and inadequate and inappropriate supplementary feeding practices [18]. Subsequent evidence shows that prolonged breastfeeding beyond 12 months, when the impact of the poor economic status of a household is begins to take effect, potentiated stunting, where children belong to poorest quintiles are more prone to stunting than others [19].

Note that the universal skilled birth attendant results in no rich–poor gaps of access to safe birthing, and coverage among the poorest and richest quintiles are 98% and 100%. The nearer the universal coverage, the smaller the rich-poor gaps; this reflected UHC achieves both health gains and nullify social inequity.

Vietnam (Figure 3b) had achieved a high coverage of all dimensions except the early initiation of breastfeeding. Vietnam, with its high level of economic growth, is the target of breast-milk substitutes (BMS) aggressive marketing, where 74% of Vietnamese mothers recalled the advertising and promotion of BMS in television, 63% of healthcare facilities and professional recalled contacts made by company representatives, and there is a 45% prevalence of point of sale with the promotion of BMS products [20].

The national average of skilled birth attendant rates was 93.8% in 2014, resulting in a rich-poor gap; the coverage among the poorest and richest quintiles is 73% and 100%, respectively. Vietnam has a strong primary healthcare service infrastructure at commune level, for which scaling up skill birth attendance seems feasible though need to strengthen the financing of health services given its high level of population coverage (87% of population in 2018).

Indonesia, see Figure 3c, had performed well for most of the nine dimensions, except access to modern methods of family planning and the early initiation of breastfeeding. A study [21] reported that 20% of Indonesian women had received advice and information on the use of BMS, 72% of women had seen BMS promotional materials, 15% received free samples and 16% received gifts. Furthermore, nearly a quarter of the health workers confirmed receiving visits from representatives of BMS companies and received gifts from the companies. Unethical market promotion was reported from Indonesia; for example, on the Wyeth Indonesia website, those who purchase the follow-up formula can join the “iPhone rally” to get the prize of a free iPhone [22]. 

Timor Leste, in Figure 3d, had a high coverage on antenatal care, early initiation of breast feeding, care seeking for pneumonia and the use of improve drinking water sources. However, there is a large gap on coverage of modern methods of family planning, skilled birth attendance, and improved sanitation.

The Philippines, in Figure 3e, despite the good coverage of antenatal care, skilled birth attendance, DTP3 coverage, improved drinking water, and birth registration, there is a low coverage of family planning. The Philippine Congress enacted Republic Act No. 10354 on Responsible Parenthood and Reproductive Health on December 18, 2012, after decades of “bitter public controversy and political wrangling” [23].

Cambodian performance on MNCH service coverage, Figure 3f, was uneven, and there was very high coverage of DTP3 and the skilled birth attendance rate, though the coverage of modern family planning, care seeking for children with pneumonia, the use of improved drinking water, and the use of improved sanitation need significant improvement.

The early initiation of breastfeeding needs improvement; Cambodia is one of the South East Asian countries that is the target of the aggressive marketing promotion of breast-milk substitute industries. Analysis of DHS showed the greatest increase in breast-milk substitute consumption through bottle feeding among urban poor, from 5.8% in 2005 to 21.7% in 2010 [24]. Despite prohibition without specific approval by the national government, breast-milk substitute companies pervasively promoted their products on television and at points of sale [25]. The average of skilled birth attendance rate was 89% in 2014. The rich-poor gap was substantial; the coverage among the poorest and richest was 75% and 98%, respectively.

The performance of MNCH service coverage in Myanmar, Figure 3g, was good on the coverage of skilled birth attendance, improved drinking water sources, and birth registration. However, the modern method of family planning, antenatal care, early initiation of breastfeeding, DTP3 coverage, and access to improved sanitation require significant improvement. 

The performance of MNCH service coverage in Lao PDR, see Figure 3h, was high in improved drinking sources of water and sanitation and birth registration. Coverage of the modern method of family planning, early initiation of breastfeeding, and care seeking for children with pneumonia were major challenges requiring policy attention. Contraceptive prevalence using the modern family planning method was as low as 49%.

Similarly, Lao is the target of BMS marketing promotion. Nestle representatives in Vientiane hospitals provided gifts, trips, and other incentives to Lao doctors and nurses, as well as marketing formula and non-formula products with the logo of a baby bear held in the breastfeeding position [26].

### 3.3. Inequity Analysis: The National Average Conceals the Inequity Gap

In the UNGA resolution, A/RES/71/313, E/CN.3/2018/2 on SDGs, it clearly urges UN Member States to produce SDG indicators disaggregated, where relevant, by income, gender, age, race, ethnicity, migratory status, disability and geographic location, or other characteristics [27].

Gaps analysis offers an evidence-informed policy decision that government health investment needs to take into account whether the poor can access care adequately; a pro-poor design of UHC is a critical aspect of ensuring that the rural population, minorities and other people living in vulnerable conditions have equitable access to care (ensuring that no one is left behind).

Detailed analysis combining two independent socio-economic parameters: domicile (urban and rural) and wealth quintiles disclosed significant inequity in Lao PDR, see Figure 4. For example, the rural poorest had a significantly lower rate of antenatal care, skill birth attendants, care seeking behavior for children with symptoms of pneumonia and access to improved sanitation than the urban poorest, except the higher rate of initiation of breastfeeding among the rural poor. Being among the rural poor was the extreme vulnerability characteristic that health policy needs to cater for this group (let alone being an ethnic minority and/or an individual living in hard-to-reach areas in which the MICS did not capture these independent parameters).

Evidence demonstrated that being a rural or urban resident does not matter so long as they belong to richest quintiles. There was no difference in MNCH service coverage across interventions; however, the coverage of family planning among the richest quintiles was lower than in the poorest quantiles.

Clearly, the rural richest were much better off than the rural poorest group on MNCH service coverage, and, likewise, the urban richest were much better off than the urban poorest. Despite no barriers on supply side in urban areas, the richest had significantly more financial access to care than the poorest. Despite the supply side limitation in rural areas, the rural richest quintiles had financial means and transport capacity to travel and seek care in the urban areas.

Combining two socio-economic parameters, domicile and wealth quintiles in Thailand in Figure 5 did not show much inequity. For example, there was a comparable coverage of nine MNCH services between the poorest quintiles in rural and urban areas, and there was comparable coverage of rural people who were poorest and richest.

This reflects the equitable distribution of health services and financial access to services by all people in rural areas—rich or poor. Rural coverage of either the rich or poor is on par with their counterparts in urban areas, though the richest urban population had a lower coverage of family planning and initiation of breast feeding.

The equitable coverage stratified by wealth quintiles and geographical areas in Thailand was the result of extensive distributions of quality services provided by district health systems. In each district with a 50,000 catchment population, there is a district hospital range from 30–60 beds, fully equipped to provide primary and secondary care. In a district, there is a network of 10–12 sub-district health centers, each center covers 5000 catchment populations that work closely with the district hospital [28]. This forms a district health system that is the contractor provider for all outpatient care, prevention and health promotion services under the Universal Coverage Scheme [29].

### 3.4. UHC Status 

Two explicit goals of UHC are improving equitable access to health services and financial risk protection. The lack of financial risk protection is reflected by the high level of out-of-pocket payment, see Figure 6.

Myanmar and Cambodia had the highest out-of-pocket health spending as a proportion of current health expenditure in 2015, hence the high incidence of catastrophic health spending. In contrast, Thailand had invested in health systems’ development for the last three decades [31], and a comprehensive range of benefit packages were offered free at the point of services, which resulted in a high level of UHC coverage index and a low incidence of catastrophic health spending.

Evidence shows that out-of-pocket payment as % of current health expenditure decreases when the share of government health spending to GDP increases [32], although there is considerable cross-country variation. Higher government spending on the health of the population improves financial risk protection and reduces the proportion of out-of-pocket payment by the household. There is a steep reduction in out-of-pocket payment when government spends 2–3% of its GDP on health, see Figure 7.

This evidence clearly generates a policy message that increased health spending by the government can reduce out-of-pocket payment, and the low level of out-of-pocket payment results in a low level of prevalence of catastrophic health spending, which is the SDG target 3.8.2.

The incidence of catastrophic health spending, using the 10% threshold in Table 1, ranged from 3.0% of households in Lao PDR to 9.8% in Vietnam. However, the low incidence of catastrophic health spending should be monitored against the prevalence of unmet needs. If the poor households decided not to use health services because of unaffordability, there is no out-of-pocket payment and no catastrophic health spending, but there is a huge welfare loss, such as disability or mortality. However, with the exception of Thailand [33], only Organization for Economic Co-operation and Development (OECD) had monitored unmet healthcare needs [34].

For monitoring the progress of SDG target 3.8.1, there is a need to review and improve the national data base that supports monitoring of these sixteen UHC services coverage [35].

## 4. Discussion

This paper found a significant achievement in U5MR and IMR reduction in selected countries in the South East Asia region, during the MDGs era, though efforts to accelerate achievement of SDG 3.2 targets in some countries by 2030 are needed. This can be done through accelerating the progress of MNCH to protect health of children and women.

One key common challenge, which emerged in all these countries, was the low level of the early initiation of breastfeeding. This challenge resulted from the commercial determinants of BMS and the industry’s aggressive and unethical marketing promotion, by violating the International Code of Marketing of Breast-milk Substitute and national law in certain countries.

To counteract the BMS industry, firstly, governments need to control advertisement and point-of-sale promotions. Traditional and online media monitoring showed high levels of point-of-sales promotions and widespread advertising. Secondly, the government needs to control the corporate donations and make contact with the target population. Studies found that companies donated a significant amount of branded equipment and promotional materials to healthcare facilities [2,36]. The regular monitoring of violation and taking legal actions for the violators are recommended. The increased prevalence of cesarean section is another reason, particularly Thailand, that contributed to the failure of the early initiation of breastfeeding [37].

In UHC status, in Figure 6, countries were classified into four quadrants using the global average. Countries with high coverage but also a high incidence of catastrophic spending need to improve their financial risk protection. Countries with low coverage and a high incidence of catastrophic spending need to boost service provision capacity, as well as expand financial protection. Countries with low coverage and a low incidence of catastrophic spending were the poor performers. In these countries, poor households who cannot afford to pay for health services may forego required care and instead choose to die at home.

Primary health care (PHC) is an overall approach that encompasses the following three aspects: (a) multisectoral policy and action to address the broader determinants of health; (b) empowering individuals, families and communities; and (c) meeting people’s essential health needs throughout their lives. Note that primary care is a subset of PHC and refers to essential, first-contact care provided in a community setting.

The PHC approach means working with multidisciplinary teams—doctors, nurses, caregivers, therapists, and others—to treat the person rather than the disease, provide health care services throughout a person’s life of a whole range of prevention, health promotion, rehabilitation, palliation services and end-of-life care and support. Such an approach allows people to develop long-term partnerships with their care providers in their communities. PHC is the foundation of a strong health system that leads to more equitable health and greater patient and health worker satisfaction.

Despite PHC’s critical role, it has been neglected in many countries in favor of a disease-specific and hospital-oriented approach, probably due to the lack of political will and underinvestment in infrastructure and the health workforce, which ensures rural retention. Political will has advanced greatly with the adoption of the Declaration of Astana [38]. Thailand health systems with a PHC-based foundation are the foundation of UHC implementation; it had proved favorable UHC outcomes in terms of improved clinical outcomes, increased efficiency, better quality of care and enhanced patient satisfaction, and financial risk protection [28].

At PHC level, Thailand offers universal access to MNCH benefit packages since pre-conception until children grow up to 5 years. Health services cover the pre-conception, ante-partum, intra-partum and post-partum periods. At the pre-conception period, services cover condom distribution at health facilities, youth-friendly clinics and outreach clinics, the promotion for Fero-Folic supplements for expectant mothers, family planning and health check-ups for couples are provided. The ante-partum period covers screening (e.g., Thalassemia, HIV); treatment (folate, iron, iodine, dental care); and health education (parental school and MNCH handbook). The intra-partum period covers delivery by skilled birth attendance and health facility with obstetric emergencies equipment. The post-partum period covers congenital hypothyroid and Phenylketonuria (PKU) screening for newborn, early and follow-up health check-up, standard “well child clinic”, free Expanded Program of Immunization (EPI), child development check-up, and counselling for breastfeeding and child development.

Pregnant women receive the MNCH handbook or “pink book” when they access an antenatal clinic for the first time. Each child receives regular check-ups according to the National Developmental Screening Program Manual (DSPM) from health personnel at 9, 18, 30, and 42 months of age at well child clinics, enabling parents to promote their child’s development at home.

Though this review offered new insights into how the health financing system and UHC contributed to the MNCH achievements, some limitations were identified. Firstly, the MNCH dataset of each country was extracted from different surveys, either a MICS or DHS survey, from different years. This hampered accuracy across country comparisons. Secondly, several factors that could contribute to MNCH outcomes, such as maternal education, women employment status and the household decision-making processes, were outside the scope of this review. Thirdly, space did not allow reviews of inter-sectoral actions for health, which are equally important, such as civil registration systems for registering every birth, which ensures the right to health of every child, early childhood development, education and female empowerment. All these factors are of equal importance as contributions by health systems and UHC. Finally, this review covered eight selected countries and could not represent the whole of the South East Asia region.

## 5. Conclusions

This paper reviewed the status of MNCH service coverage in relation to the UHC status of eight selected middle-income countries in the South East Asia region that shared a similar socio-cultural background. This paper contributed to understanding how UHC facilitates access to MNCH services. Though most countries have made progress in MNCH outcomes, progress towards UHC requires further policy attention. This paper suggested a few policy recommendations.

Firstly, increase government spending on health and favorable benefit packages, including MNCH services, can facilitate access to service and reduce out-of-pocket payment. The low level of out-of-pocket payment results in a low level of prevalence of catastrophic health spending, which is the SDG target 3.8.2. Policy goals are to increase government fiscal space as measured by tax as % GDP, through tax reform and effectiveness in revenue collection by the Finance Ministry, and increase fiscal space for health through increasing the proportion of general government spending on health as % of general government expenditure through sustained political leadership and funding commitment. Reliance of tax revenue is more sustainable than non-tax revenue, which can be interfered by the fluctuation of the global market price of natural resources, such as oil price and minerals.

Secondly, most countries in the region are no longer eligible for donors’ funding when transitioning from low- to lower-middle income countries. There is a need to exert political leadership at the highest level of government in mobilizing domestic funding replacing donors’ source in order to prevent the interruption of essential programs previously funded by donors.

The third recommendation is to strengthen PHC and safeguard the poor, mothers and children with a comprehensive benefit packages as a solid platform for implementing UHC. PHC, due to its geographical proximity, facilitates better access to MNCH services.

The final recommendation is to sustain regular household surveys. In particular, MICS, DHS and others, such as income and expenditure surveys, are critical for monitoring progresses towards all health-related SDG and equity analysis such as urban-rural, rich-poor and educational gaps.

The future work may assess the competency of PHC and hospitals in managing perinatal services, which can minimize the preventable maternal and neonatal mortality, which are the two SDG indicators. These are the remaining gaps in some countries.

## Figures and Tables

**Figure 1 ijerph-17-03366-f001:**
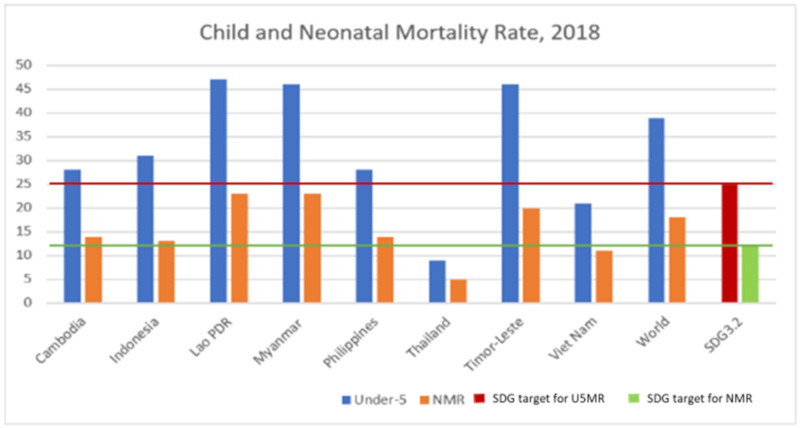
Child and neonatal mortality per 1000 live births in 2018, selected countries in the South East Asia region. Data source: Levels and trends in child mortality 2019, UNICEF, WHO, World Bank Group and United Nations [11].

**Figure 2 ijerph-17-03366-f002:**
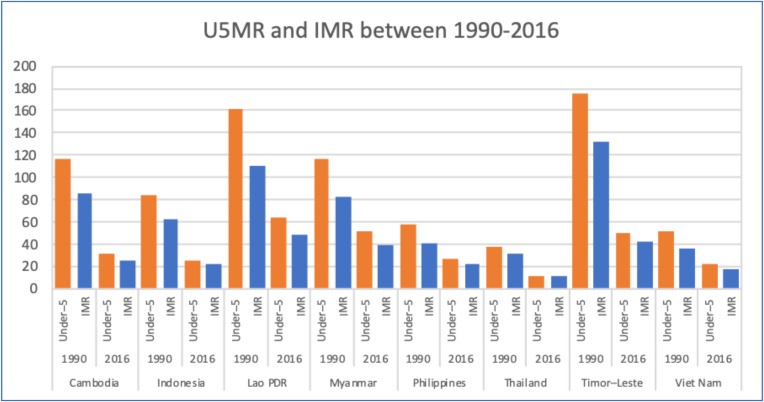
Under-5 mortality and infant mortality rate in 1990 and 2016 of selected countries in the South East Asia region. Data source: Levels and trends in child mortality 2019, UNICEF, WHO, World Bank Group and United Nations [11].

**Figure 3 ijerph-17-03366-f003:**
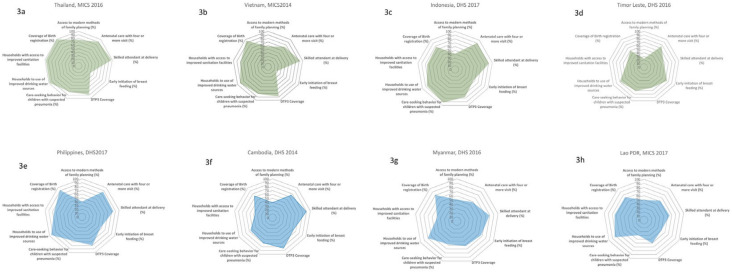
Achievement of women and children protection based on nine dimensions of maternal, newborn, and child health (MNCH) service coverage in eight selected countries. Data source: Thailand Multi-Indicator Cluster Survey (MICS) 2016, Viet Nam MICS 2014, Indonesia DHS 2017, Timor Leste DHS 2016, Philippines DHS 2017, Cambodia DHS 2014, Myanmar DHS 2016, and Lao PDR MICS 2017 (data from the reports of MICS and DHS in South East Asian countries [14]. (**a**) family planning (modern methods); (**b**) four quality antenatal care (ANC) visits; (**c**) skilled birth attendant rates; (**d**) the early initiation of breastfeeding; (**e**) third dose of Diphtheria-tetanus-pertussis (DTP3) immunization coverage; (**f**) care seeking for children with pneumonia; (**g**) the use of improved drinking water sources; (**h**) the use of improved sanitation; and coverage of birth registration.

**Figure 4 ijerph-17-03366-f004:**
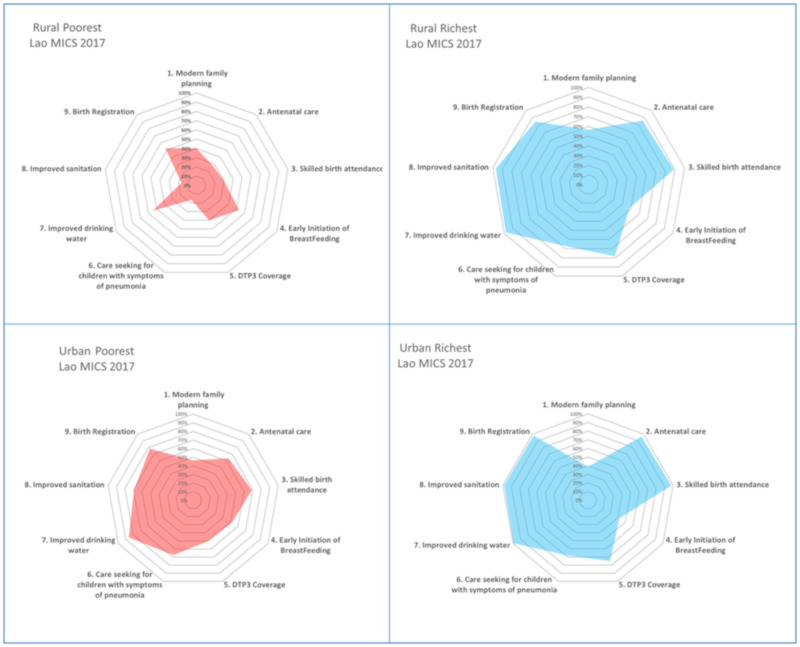
Synergies of vulnerability: urban/rural and rich-poor quintiles, Lao PDR. Data source: Lao PDR MICS 2017 [14].

**Figure 5 ijerph-17-03366-f005:**
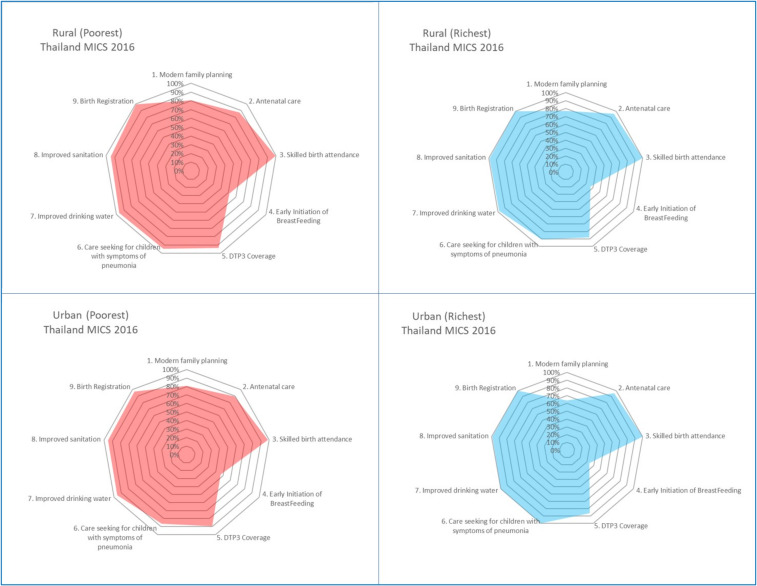
Synergies of vulnerability: urban/rural and rich-poor quintiles, Thailand. Data source: Thailand MICS 2016 [14].

**Figure 6 ijerph-17-03366-f006:**
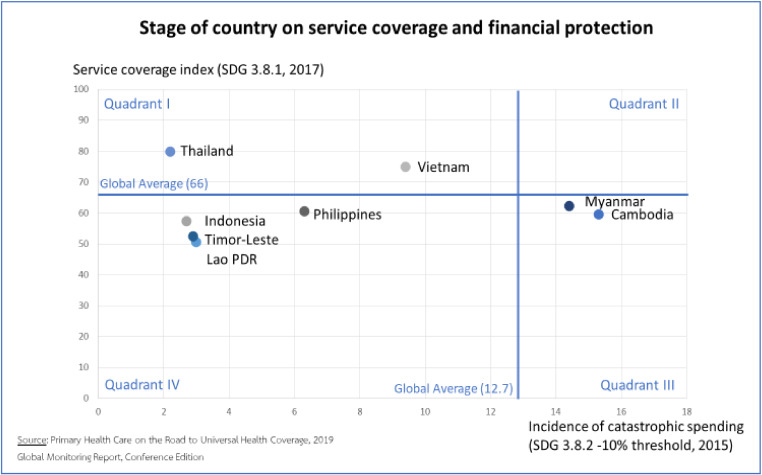
Stage of country on service coverage and financial protection. Data source: Primary Health Care on the Road to Universal Health Coverage, 2019: Global Monitoring Report, Conference Edition [30].

**Figure 7 ijerph-17-03366-f007:**
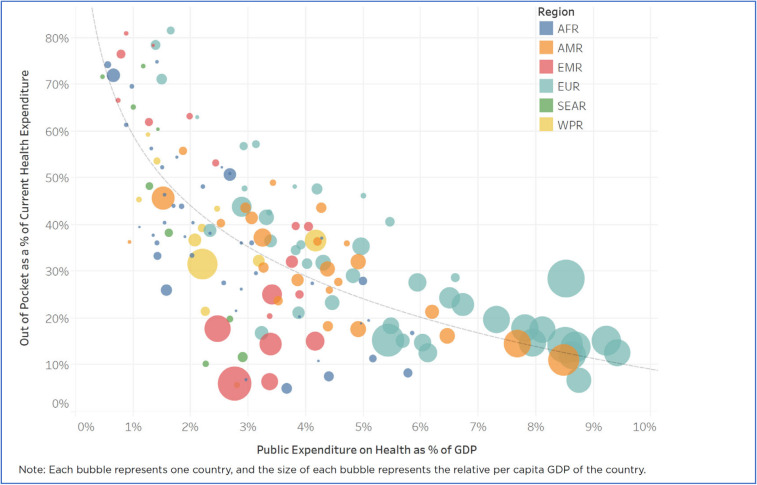
Negative correlation between public health spending as % GDP and the proportion of out-of-pocket payment as % total health expenditure. Data source: New Perspective on Global Health Spending for Universal Health Coverage, Global report: Conference copy for consultation, WHO, 2018 [32].

**Table 1 ijerph-17-03366-t001:** Countries at a glance.

Key Parameters	Cambodia	Indonesia	Lao PDR	Myanmar	Philippines	Timor Leste	Thailand	Vietnam
I. Demography								
• Population, total (millions)	16.2	267.7	7.1	53.7	106.7	1.27	69.4	95.5
II. Economic and fiscal space								
1. GDP per capita (current US$), 2018	1512	3894	2568	1326	3103	2036	7274	2564
2. GDP growth (annual %), 2018	7.5	5.2	6.5	6.2	6.2	2.8	4.1	7.1
3. Revenue, excluding grants (% of GDP), 2017	18.6	12.2	NA	15.8	15.6	38.4	19.2	21.8 (2013)
4. Tax revenue (% of GDP), 2017	15.8	9.9	NA	6.0	14.2	12.9	14.8	19.1 (2013)
III. Fiscal space for health								
5. Current health expenditure per capita (current US$), 2016	78	112	55	62	129	80	222	123
6. Current health expenditure (% of GDP), 2016	6.1	3.1	2.4	5.1	4.4	4.0	3.7	5.7
7. Domestic general government health expenditure (% of CHE), 2016	21.8	44.7	32.4	20.1	31.5	55.8	78.1	47.4
8. Domestic general government health expenditure (% of GGE), 2016	6.2	8.3	3.7	4.8	7.1	3.2	15.3	8.9
9. External health expenditure (% of CHE), 2016	18.9	0.4	18.1	5.9	2.2	31.6	0.2	2.3
10. Out-of-pocket expenditure (% of CHE), 2016	58.6	37.3	46.4	74.0	53.9	8.9	12.1	44.6
IV. UHC indicators								
11. UHC coverage index, 2015	55	49	48	NA	58	47	75	73
12. Catastrophic spending, >10% household spending, % households *	NA	3.6 (2015)	3.0 (2007)	NA	6.3 (2015)	NA	3.4 (2010)	9.8 (2014)
13. Catastrophic spending, >25% household spending, % households *	NA	0.4 (2015)	0.3 (2007)	NA	1.4 (2015)	NA	0.7 (2010)	2.1 (2014)

Data Source: The World Bank https://data.worldbank.org/ [9]. Note * World Health Organization http://apps.who.int/gho/portal/uhc-financial-protection-v3.jsp [10].

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
