# Peer review of "Children and Women’s Health in South East Asia: Gap Analysis and Solutions"

_ijerph, 2020, doi:10.3390/ijerph17103366_

Round 1
Reviewer 1 Report
It is valuable review about children and women health in East Asia and Pacific. For those countries, social-economic development is going-on and improvement of maternal and child health care has been found. Authors analyzed the gap in maternal and child health care compared with MDG and SDG commitment, and also provided some suggestion on solutions. However, the following issues should be addressed further.
- Authors said that “The selection of these eight countries is based on availability of comparable statistics mainly drawn from the Multi-Cluster Indicators Surveys or Demographic and Health Surveys.” But it seemed unclear about why to select those countries. As we know, there are lots of countries in East Asia and Pacific. Why was Malaysia not included? Maybe countries included here could share something in common like economic level, culture background or something like that. Therefore, those countries included here could not reflect general situation of East Asia and Pacific. I suggest authors should provide a clear rationality on selection of countries.
- About solutions, author mentioned some solutions in both the part of discussion and conclusion. It seemed not be focused. Maybe authors present clear point of solution to main issue after discussion. But for conclusion, author should provide clear and concrete conclusion, not repeat discussion.
- Authors should provide clear limitation about this review, although I found something in the conclusion. They should be presented in the part of discussion.
Author Response
Dear Reviewer 1, please see the attachment.

Reviewer 2 Report
- The manuscript, as written, does not present a coherent project.
- A more comprehensive literature review is required to provide the context for the research described and the findings reported.
- A more detailed description of the data collection methods used and its limitations is needed.
- More information is needed to describe the data analysis that was conducted.
- How the stated findings contribute to progressing this research area are unclear.
- What are the limitations of this research study?
7. What is the direction for future work?
- English grammar needs corrections.
- As written, the manuscript reads as a draft.
Author Response
Dear Reviewer 2, please see the attachment.

Round 2
Reviewer 1 Report
Authors has addessed most of my comments, and the manuscipt has been improved much.
Reviewer 2 Report
Significant improvement!